# Association between Low-Density Lipoprotein Cholesterol and Vascular Biomarkers in Primary Prevention

**DOI:** 10.3390/biomedicines11061753

**Published:** 2023-06-18

**Authors:** Michaela Kozakova, Carmela Morizzo, Giuli Jamagidze, Daniele Della Latta, Sara Chiappino, Dante Chiappino, Carlo Palombo

**Affiliations:** 1Department of Clinical and Experimental Medicine, University of Pisa, 56126 Pisa, Italy; michaela.kozakova@esaote.com; 2Esaote SpA, 16152 Genova, Italy; 3Department of Surgical, Medical and Molecular Pathology and Critical Care Medicine, School of Medicine, University of Pisa, 56126 Pisa, Italy; carmela.morizzo@unipi.it; 4Imaging Department, Fondazione Toscana G. Monasterio, 54100 Massa, Italy; giulisana@yahoo.com (G.J.); ddellalatta@terarecon.com (D.D.L.); sara.chiappino@gmail.com (S.C.); dchiappino@gmail.com (D.C.); 5Bioengineering and Deep Health Units, Fondazione Toscana G. Monasterio, 54100 Massa, Italy

**Keywords:** lipid-lowering treatment, primary prevention, coronary calcium, intima-media thickness, arterial stiffness, carotid artery, femoral artery

## Abstract

Several noninvasive vascular biomarkers have been proposed to improve risk stratification for atherothrombotic events. To identify biomarkers suitable for detecting intermediate-risk individuals who might benefit from lipid-lowering treatment in primary prevention, the present study tested the association of plasma LDL-cholesterol with coronary artery calcification (CAC) Agatston score, high carotid and femoral intima-media thickness (IMT), low carotid distensibility and high carotid-femoral pulse-wave velocity in 260 asymptomatic individuals at intermediate cardiovascular risk and without diabetes and lipid-lowering treatment. High or low vascular biomarkers were considered when their value was above the 95th or below the 5th percentile, respectively, of the distribution in the healthy or in the study population. LDL-cholesterol was independently associated with the CAC score = 0 (OR 0.67; 95%CI 0.48–0.92, *p* = 0.01), CAC score > 100 (1.59; 1.08–2.39, *p* = 0.01) and high common femoral artery (CFA) IMT (1.89; 1.19–3.06, *p* < 0.01), but not with other biomarkers. Our data confirm that in individuals at intermediate risk, lipid-lowering treatment can be avoided in the presence of a CAC score = 0, while it should be used with a CAC score > 100. CFA IMT could represent a useful biomarker for decisions regarding lipid-lowering treatment. However, sex- and age-specific reference values should be established in a large healthy population.

## 1. Introduction

Low-density lipoprotein cholesterol (LDL-C) is the most abundant atherogenic lipoprotein in plasma, and its infiltration of the arterial wall is considered a key event in the initiation and progression of the atherosclerotic process. Increased plasma LDL-C levels are causally related to atherosclerotic cardiovascular (CV) disease [1] and lowering LDL-C values by lifestyle or therapeutic interventions has been shown to reduce the risk of CV events, both in primary and secondary prevention [1,2,3,4,5]. 

The 2019 ACC/AHA Guideline on the Primary Prevention of Cardiovascular Disease [6] suggests statin treatment not only in individuals with diabetes, very high LDL-C levels (≥190 mg/dL) or high 10-year risk of CV disease (≥20%), but even in individuals at intermediate CV risk (≥7.5–20%), in the presence of risk enhancers such as coronary artery calcification (CAC). In the absence of CAC, statin therapy can be avoided or withdrawn, while in the presence of the Agatston CAC score above 100, statins are recommended at any age. 

CAC, a highly specific feature of coronary atherosclerosis, is the most predictive single CV risk marker in primary prevention. CAC scoring using the Agatston method was successfully incorporated into the coronary heart disease prediction model based on traditional risk factors [7,8,9,10]. However, the utility of the Agatston score for evaluating the effectiveness of statin therapy is limited because the score considers both the total area and the maximal density of coronary calcification, and as demonstrated by serial intravascular ultrasound, intensive statin therapy induces regression of atheroma volume but promotes its calcification [11,12]. This process leads to the stabilization of the atherosclerotic lesion but may increase the CAC score. Various studies have shown that statin treatment can accelerate coronary calcification [13,14], even as it reduces all-cause mortality and major vascular events among people without evidence of CV disease [15]. Therefore, the CAC score helps define individual CV risk. Still, repeated computed tomography to assess the response to statin therapy is probably inappropriate, considering also the cumulative radiation dose [16]. 

Other noninvasive vascular biomarkers, such as carotid and femoral intima-media thickness (IMT) and plaques or carotid and aortic stiffness, have been proposed for risk estimation and decision-making regarding statin treatment, as well as for assessing response to statin therapy. Previous studies have demonstrated that statins can slow the progression of carotid IMT and improve arterial stiffness [17,18,19,20], but this beneficial effect is not necessarily related to the main action of statins, i.e., lowering serum LDL-C levels. Statins have pleiotropic effects that include reducing inflammatory cytokines and reactive oxygen species, inhibiting smooth muscle cell proliferation, improving endothelial function and lowering blood pressure (BP) [21,22,23,24,25]. These mechanisms may be involved in reducing arterial wall thickness and stiffness. 

The aim of this cross-sectional study was to evaluate the association of various noninvasive vascular biomarkers [26,27] with plasma LDL-C levels in order to identify biomarkers that could reflect the impact of LDL-C on vascular trees and could be used to select individuals suitable for lipid-lowering treatment and to monitor its effect. The study population included asymptomatic individuals at intermediate 10-year risk of CV disease, without diabetes and with LDL-C ≥ 190 mg/dL, i.e., individuals for whom additional risk assessment is recommended before starting statin therapy [6]. Vascular biomarkers tested were CAC score, high common carotid artery (CCA) or common femoral artery (CFA) IMT, low CCA distension and high carotid-femoral pulse wave velocity (cfPWV). Abnormally high or low vascular biomarkers were considered when their value was above the 95th percentile or below the 5th percentile [28], respectively, of distribution in a healthy population (if data in a healthy population were available) or in the study population (if data in healthy population were not available). 

## 2. Materials and Methods

### 2.1. Study Population and Protocol

The study population is a part of the population enrolled in the prospective cohort study “MHeLP, Montignoso Heart-Lung Project”, aimed at defining the predictive value of CAC score for CV events in a community-based (the village of Montignoso, Tuscany, Italy) general population. The original population consisted of 638 individuals. For this study, we included only individuals aged 40–75 years, without CV symptoms, lipid-lowering therapy, and diabetes, with plasma LDL-C levels < 190 mg/dL and at intermediate 10-year risk of CV disease as estimated by Framingham Risk Score. The final population consists of 260 individuals.

All individuals underwent an examination protocol that included anthropometry, brachial BP measurements, a fasting blood test, ECG, a high-resolution carotid and femoral ultrasound and a computed tomography scan. Carotid-femoral pulse wave velocity was measured in 182 individuals. Hypertension was defined as systolic BP > 140 mmHg and/or diastolic BP  > 90 mmHg or hypertensive treatment [29].

### 2.2. Body Size and BP Measurement

Body weight and height were measured, and body mass index (BMI) was calculated. Waist circumference was measured as the narrowest circumference between the lower rib margin and anterior superior iliac crest. Brachial BP was measured at two visits by a validated digital electronic tensiometer (Omron, model 705cp, Kyoto, Japan) in participants seated for at least 10 min, using regular or large adult cuffs according to the arm circumference. Two measurements were taken at both visits, separated by 2-min intervals, and the average was calculated. The average of two separate visits was used to estimate BP (mmHg). Pulse pressure was calculated as the difference between systolic and diastolic BP.

### 2.3. Assessment of the 10-Year Risk of CV Disease 

A 10-year risk of CV disease was estimated by the Framingham risk score prediction model that considers age, total cholesterol, high-density lipoprotein cholesterol (HDL-C), systolic brachial BP, ongoing treatment of hypertension, smoking and diabetes status [30]. The risk was classified as intermediate when the 10-year risk of CV disease ranged between 7.5 and 20% [6]. 

### 2.4. Assessment of LDL-C Plasma Level 

Plasma LDL-C was measured by direct enzymatic colorimetric method (Beckman Coulter, Danaher Corporation, Sunnyvale, CA, USA). LDL-C levels were considered as optimal (<100 mg/dL), near-optimal (100–129 mg/dL), borderline high (130–159 mg/dL) and high (160–189 mg/dL) as recommended [31]. 

### 2.5. Coronary Calcium Score (CAC)

A low-dose radiation scan without contrast agent was done (120 KV, 60 mA) by a 64-detector scanner (Aquilion 64; Toshiba Medical Systems, Otawara, Japan) (2–3 mSv estimated dosimetry, with 1- and 3-mm collimation and reconstruction thickness, respectively). Prospective electrocardiographic triggering in sequential slice mode was used for scanning the heart. The CAC score was calculated by Agatston [32] with a dedicated program available (Vitrea 2.0; Vital Images Inc., Minnetonka, MN, USA). The Agatston score was determined by multiplying the area of calcification expressed in mm^3^ by the corresponding density number using the following density scale (1 = 130–199 Hounsfield units (HU), 2 = 200–299 HU, 3 = 300–399 HU, 4 = ≥400 HU). For this study, we identified individuals with a CAC score = 0 and a CAC score > 100.

### 2.6. Vascular Examination

All vascular examinations were carried out by the same operator (G.J.) in a quiet room with a stable temperature of 22 °C on individuals resting comfortably for at least 15 min in the supine position. All individuals were asked to abstain from cigarette smoking, caffeine and alcohol consumption and vigorous physical activity for 24 h. 

Carotid ultrasound was performed on the right CCA using an ultrasound scanner equipped with a 10 MHz linear probe (MyLab 70, Esaote, Genova, Italy) and implemented with a previously validated radiofrequency-based tracking of the arterial wall (QIMT^®^, QAS^®^, Esaote, Genova, Italy) that allows an automatic and real-time determination of far-wall CCA IMT, CCA outer diameter and distension with a high spatial and temporal resolution (sampling rate of 550 Hz on 32 lines). CCA IMT, diameter and distension were measured within a rectangular 1-cm-long ROI placed approximately 1 cm before the flow divider. IMT was defined as the distance between the lumen-intima and media-adventitia interfaces of the far (posterior) wall. Distensibility coefficient (DC) was calculated from the distension curves as follows: DC = (∆A/A)/PP, where A = π ∗ (D/2)^2^, ∆A = π ∗ [(D + ∆D)/2]^2—^π ∗ (D/2)^2^, D = diastolic outer diameter and PP = Pulse Pressure [33].

Femoral ultrasound was performed on the right CFA using the same ultrasound scanner with radiofrequency-based tracking of the arterial wall. Far wall IMT was measured within a rectangular 1-cm-long ROI placed approximately 1 cm before the flow divider. 

To identify individuals with high CCA IMT, the sex- and age-specific normality tables deriving from IMT measurements performed by the same radiofrequency-based system in 4234 healthy men and women were used [34], and a cutoff point of the 95th percentile for given sex and age was adopted [28]. To identify individuals with low CCA DC, the sex- and age-specific normality tables deriving from DC measurements performed by the same radiofrequency-based system in 3601 healthy men and women were used [33], and a cutoff point of the 5th percentile for given sex and age was adopted [28]. As no normalcy data were published for CFA IMT, a cutoff point of the 95th percentile of CFA IMT distribution in our population was adopted for defining high CFA IMT. 

Radiofrequency-derived measures represent an average of over six consecutive cardiac beats. The mean of two acquisitions was used for statistical analysis. BP was measured at the left brachial artery (Omron, Kyoto, Japan) during each acquisition of the distension curves. Intra-individual variability of acquisitions was evaluated in 25 volunteers, in whom the acquisitions were performed in two sessions separated by 30 min. Brachial pulse pressure was comparable between the two acquisitions (*p* = 0.88). Intra-individual variability of CCA IMT, CCA distension and CFA IMT was 6.7 ± 4.2, 8.7 ± 6.4% and 8.1 ± 5.4%, respectively.

Carotid-femoral pulse wave velocity (cfPWV) was measured in 182 participants according to current guidelines [35] using the Complior device (Alam Medical, Vincennes, France). To identify individuals with high cfPWV, the sex-, age- and mean BP-specific normality tables deriving from cfPWV measurements performed in 2158 healthy men and women were used [36] and a cutoff point of the 95th percentile for given sex, age and mean BP was adopted [28]. Intra-individual variability of carotid-femoral PWV measurement was 4.3 ± 2.8%.

### 2.7. Statistical Analysis

Data are expressed as mean ± SD, and categorical data as percentages. Variables with skewed distribution were summarized as median [interquartile range] and were logarithmically transformed for parametric statistical analysis. Multivariate logistic regression was performed to identify risk factors associated with CAC score = 0, CCA score > 100, high CCA IMT, high CFA IMT, low CCA DC and high cfPWV. Results are given as odds ratio (OR) and 95% confidence interval (CI). ORs were calculated for 1SD of the continuous variable. Statistical tests were two-sided, and significance was set at a value of *p* < 0.05. 

## 3. Results

Characteristics of the study population are reported in Table 1. Twenty-nine individuals had optimal LDL-C levels, ninety-three near optimal levels, ninety-five borderline high and forty-three high levels. 

Table 2 reports data on CAC score, CCA IMT and distensibility, CFA IMT and cfPWV. One hundred eighty-two individuals had CAC score = 0, 36 had CAC score > 100, 49 had high CCA IMT, 15 had low CCA DC, 25 had high CFA IMT, and 27 had high cfPWV.

Figure 1 reports risk factors associated with CAC score = 0, CAC score > 100, high CCA IMT or CFA IMT, low CCA DC and high cfPWV. LDL-C was independently associated with CAC score = 0, CAC score > 100 and high CFA IMT, but not with high CCA IMT, low CCA DC or high cfPWV. Neither HDL-C nor triglycerides were related to any vascular biomarker. 

Odds ratios are calculated for 1SD of the continuous variable.

## 4. Discussion

In our asymptomatic population at intermediate CV risk, LDL-C was associated with coronary calcification and CFA IMT in the highest 5 percent distribution. Neither high CCA IMT, low carotid distensibility, nor high aortic stiffness was associated with plasma lipids. 

These results confirm the association between LDL-C levels and coronary calcification in asymptomatic patients at intermediate CV risk. The 2019 ACC/AHA Guideline on the Primary Prevention of Cardiovascular Disease [6] does not recommend statin therapy in intermediate-risk individuals if the CAC score is 0, while it recommends starting statin therapy at any age if the CAC score is greater than 100. In our population, LDL-C was associated with a CAC score of 0 with OR = 0.67 and a CAC score above 100 with OR = 1.59.

The relationship between LDL-C exposure and vascular calcification has been demonstrated in both clinical and experimental studies. It reflects the fact that in the arterial wall are cells capable of osteoblastic differentiation and mineralization and that oxidized lipids accumulated in the subendothelial space induce differentiation of these cells and promote calcification [37,38]. Vascular smooth muscle cells, endothelial cells, fibroblasts, mesenchymal stem cells, and endothelial progenitor cells can transdifferentiate into osteoblast-like cells [39]. However, calcification in atherosclerotic lesions is associated with both progression of disease (microcalcification) and the healing of inflammation (macrocalcification), and statins may reduce arterial wall inflammation through a variety of mechanisms [40]. In the Heinz Nixdorf Recall Study, statin intake enhanced CAC progression, mostly in the less advanced stage of atherosclerosis, but this progression did not increase the risk for coronary events [14]. A meta-analysis of 7 studies suggested that in asymptomatic populations at high risk of CV diseases, statins do not reduce or enhance CAC score but slightly slow its progression, above all in individuals with CAC score > 400 [41]. CAC score can therefore help adjust individual CV risk and select suitable candidates for statin therapy, but its value for monitoring treatment efficacy is limited. 

Among the other vascular biomarkers tested in our study, neither high CCA IMT, low CCA distensibility, nor high aortic stiffness was related to plasma LDL-C. On the other hand, all these biomarkers were independently associated with pulse pressure and waist circumference. Pulse pressure is considered a major mechanical determinant of arterial remodeling because repeated cyclic stress exerts a fatiguing effect on the load-bearing elements of the arterial media, above all on elastin, causing its fracture and degeneration and leading to luminal enlargement and wall stiffening. Luminal enlargement is accompanied by a further increase in circumferential wall stress, which activates intracellular signaling pathways promoting smooth muscle cell proliferation and migration, resulting in wall thickening and reducing stress [42]. Indeed, the results of various studies suggest that an increase in carotid IMT reflects, rather than atherosclerosis, a physiologic remodeling aimed to maintain stable circumferential wall stress when BP increases [43,44]. One of the pleiotropic effects of statins is BP reduction [25], above all in individuals with higher BP and in users of anti-hypertensive medication [45,46]. Thus, the reported deceleration of carotid IMT progression with statin therapy [17,18,19] could be partially explained by the decrease in BP and arterial wall stress and by the inhibition of smooth muscle cell proliferation and migration [24]. In a longitudinal community-based Taiwanese study evaluating the effect of changes in LDL-C and systolic BP on carotid IMT, baseline IMT was associated with both LDL-C and BP, but changes in IMT during a 6-year observational period were associated only with changes in BP [47].

Another independent determinant of high CCA IMT, low CCA distensibility and high aortic stiffness was waist circumference. Central obesity is related to low-grade systemic inflammation [48,49], and in hypercholesterolemic patients, low-grade systemic inflammation and abdominal fat were the main determinants of reduced arterial distensibility [50]. Similarly, in patients with type 2 diabetes mellitus, carotid IMT was associated with waist circumference and plasma inflammatory markers [51]. Therefore, anti-hypertensive and anti-inflammatory action of statins might explain the positive effect of stain therapy on elastic arteries structure and function, independent of LDL-C levels [21,25]. 

Unlike CCA IMT, high CFA IMT was independently associated with plasma LDL-C. Previous studies comparing the relationships between risk factors and IMT in the carotid and femoral arteries have not reported significant differences. In the AXA study [52], evaluating the association between large-artery wall thickness and risk factors in men and women 17 to 65 years old, both carotid and femoral IMT were associated with age, BMI, BP, total cholesterol and glucose, and in a study including normocholesterolemic and hypercholesterolemic subjects, carotid and femoral IMTs increased with total cholesterol (r = 0.35, *p* < 0.001 for both arteries) and LDL-C (r = 0.33, *p* < 0.001 and r = 0.34, *p* < 0.001, respectively) in a similar way [53]. Yet, when the impact of 2-year simvastatin treatment on carotid and femoral IMT was assessed in patients with familial hypercholesterolemia, a more important IMT regression was observed in the femoral than in the carotid artery (−0.283 vs. −0.053 mm). In addition, hypercholesterolemic patients with a history of CV disease compared with those without CV disease had significantly higher CFA IMT at baseline (2.31 ± 0.81 vs. 1.72 ± 0.63 mm, *p* < 0.001), whereas differences in CCA IMT were less significant (0.99 ± 0.17 vs. 0.90 ± 0.19 mm, *p* = 0.01) [54]. Based on these results, authors suggested that the femoral artery could be more sensitive to the reduction of LDL-C levels and that a greater regression of femoral IMT might be clinically relevant because the association between IMT and CV disease was better in the femoral compared to the carotid artery. Likewise, in the Regression Growth Evaluation Study (REGRESS), a double-blind, placebo-controlled, prospective study of 885 men with coronary artery disease, the most significant effect of pravastatin was seen in the CFA IMT. During the 2-year observational period, far wall IMT in CFA increased by 0.13 ± 0.05 (SE) mm in the placebo group but decreased by 0.06 ± 0.05 mm in the pravastatin group (*p* = 0.004). In contrast, the two groups did not differ for changes in far wall CCA IMT (0.05 ± 0.02 vs. 0.04 ± 0.02 mm; *p* = 0.67) [55].

### 4.1. Study Limitations

We cannot evaluate the association between LDL-C levels and carotid or femoral plaques because the prevalence of plaques in our asymptomatic participants at intermediate CV risk was very low. C-reactive protein or other markers of systemic inflammation were not assessed.

### 4.2. Conclusions

Among vascular biomarkers used in primary prevention of asymptomatic individuals at intermediate 10-year CV risk, only coronary calcification, and CFA IMT in the highest 5 percent of distribution were independently related to plasma LDL-C levels. To identify intermediate-risk individuals who can benefit from lipid-lowering therapy and to monitor its effect, age- and sex-specific CFA IMT reference values should be established in a large healthy population, as has been done for CCA IMT, CCA DC and cfPWV. High CCA IMT, low CCA distension and high aortic stiffness were not associated with blood lipids but with pulse pressure and waist circumference. The latter observation suggests that the positive effect of statins on the elastic artery wall thickness and stiffness, already described by others, could be independent of LDL-C and mediated by the anti-hypertensive and anti-inflammatory actions of statins. Overall, our data indicate that different vascular biomarkers may reflect the impact of different risk factors on CV systems and that assessing more than one biomarker could provide a more accurate estimate of CV risk. 

## Figures and Tables

**Figure 1 biomedicines-11-01753-f001:**
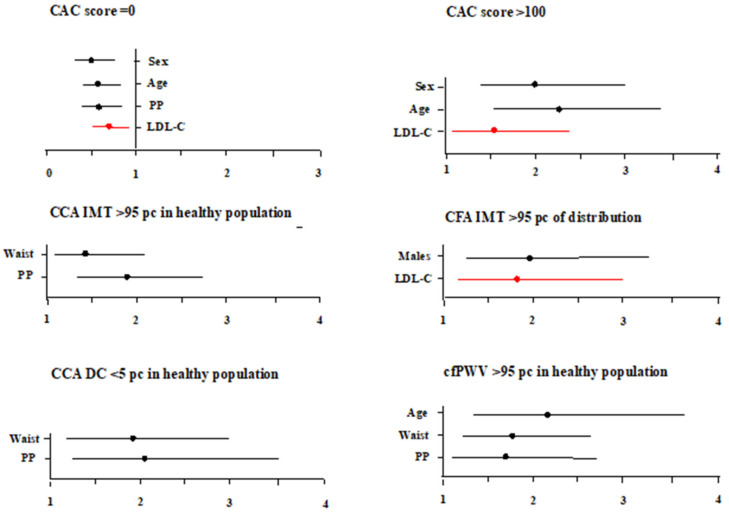
Risk factors associated with CACs score = 0, CAC score > 100, high CCA IMT, high CFA IMT, low CCA DC and high cfPWV.

**Table 1 biomedicines-11-01753-t001:** Characteristics of Study Population.

	Mean ± SD, Median [IR], *n* (%)	Range
Male: Female	119 (46):141 (54)	
Age (years)	63 ± 6	50–74
BMI (kg/m^2^)	26.9 ± 3.9	17.3–51.7
Waist circumference (cm)	95 ± 11	67–147
Systolic BP (mmHg)	136 ± 14	105–180
Pulse pressure (mmHg)	61 ± 13	30–95
Total cholesterol (mmo/L)	5.45 ± 0.78	3.28–6.88
LDL-cholesterol (mmo/L)	3.42 ± 0.6	1.66–4.85
LDL-C O:NO:BH: H	29 (11):93 (36):95 (36):43 (17)	
HDL-cholesterol (mmo/L)	1.58 ± 0.37	0.89–2.64
Triglycerides (mmo/L)	0.89 [0.75]	0.23–3.89
Fasting glucose (mmo/L)	5.36 ± 0.49	4.44–6.70
Current smoking (yes)	43 (17)	
Hypertension (yes)	113 (43)	
Hypertensive treatment (yes)	38 (15)	

O: optimal; NO: near-optimal; BH: borderline high; H: high.

**Table 2 biomedicines-11-01753-t002:** CAC Score, CCA IMT and Distensibility, CFA IMT and Aortic Stiffness.

	Mean ± SD, *n* (%)	Range
CAC = 0	182 (70)	
CAC score > 100	36 (14)	
CCA IMT (microns)	731 ± 135	459–1202
CCA IMT > 95th percentile	49 (19)	
CCA DC (10^−3^kPa^−1^)	14.2 ± 4.5	4.31–30.2
CCA DC < 5th percentile	15 (6)	
CFA IMT (microns)	736 ± 202	340–1671
CFA IMT > 95th percentile	25 (10)	
cfPWV (m/s) (*n* = 182)	9.4 ± 2.4	4.1–23.6
cfPWV > 95th percentile (*n* = 182)	27 (15)	

## Data Availability

The data presented in this study are available on request from the corresponding author. The data are not publicly available due to ethical and privacy reasons.

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
