# Peer review of "Association between Low-Density Lipoprotein Cholesterol and Vascular Biomarkers in Primary Prevention"

_biomedicines, 2023, doi:10.3390/biomedicines11061753_

Round 1
Reviewer 1 Report
Comments to the authors:
1. Standard level and the risk level of LDL should be provided.
2. The measurement of the biomarker (LDL) and assessment should also be defined in the method section.
3. The main question addressed by the research is early diagnosis of CV. The type of investigation was novel compared with other published material. The specific improvements should the authors consider regarding the methodology is biomarker assessment.
Moderate editing of English language required
Author Response
1) Standard and risk levels of LDL are provided.
see Methods, Page 6, lines 134-136, Table 1 and Reference 31
2) The assessment of LDL-cholesterol is now described in Methods.
see Methods, Page 6, lines 132-134
3) Additional relevant references were added and discussed in Introduction and Discussion section.
4) The use of English language has been revised.
Reviewer 2 Report
The study tries to identify non-invasive vascular biomarkers to improve risk stratification of atherothrombotic events.
It really is a study that is interesting, with a study of the appropriate art, a methodology, analysis of correct results.
And it is humble in the conclusions, so it can be published in the current state
Author Response
Thank you for your positive evaluation.
Reviewer 3 Report
Q1. Line 44 Please define CAC score. Is the CAC score the same as in line 46 Agatston CAC score? Please explain the Agatston CAC score
Q2 line 44 In absence of CAC, statin therapy can be avoided
In absence of CAC, statin therapy can be “stopped”?
Q3: line 60-61 smooth muscle cell proliferation, improvement of endothelial function a….
Please check that the words in these two lines are not the same size and the same types.
Q4: line 62: All these mechanisms could participate on reduction of arterial wall thickness and stiffness.
All these mechanisms may involve the reduction of the arterial wall thickness and the improvement of stiffness.
Q5 line 64-65 To identify noninvasive vascular biomarkers that could improve the selection of candidates for statin therapy in primary prevention
To search for noninvasive vascular biomarkers as potential candidates for monitoring statin therapy in primary prevention against cardiovascular diseases.
Q6 line 68-74 you said that you have identified individuals with ………. You should mention “what happen” on these individuals, or what will happen on them, or any disadvantages or advantages of them. In this way, you then can emphasize your findings. If you did not mention this, your readers may confuse about your findings.
Q7. 2.1 Study population”s” and protocol”s”
Q8 Coronary calcification may therefore help to identify candidates for statin therapy, but its value for monitoring the effectiveness of treatment is less clear.
This sentence is not clear and not easy to be understood.
Q9. Please be careful to the use of “since”
Q10. A large English revision is necessary.
A large English revision is necessary.
Author Response
Q1) The CAC score in guideline and in this study is the same. The question of Agatston score is discussed in more details.
see Introduction, Page 3, lines 62-74
Methods, Page 6, lines 144-147
Q2) In absence of CAC, statin therapy can be withdrawn.
see Introduction, Page 3, lines 63
Q3) Wrong characters have been corrected.
Q4) The sentence regarding mechanisms has been rephrased.
see Introduction, Page 4, lines 86-87
Q5) and Q6) This part has been rewritten also because we want to underline that our study is a cross-sectional study and thus no data on events are available.
see Introduction, Page 4, lines 88-91
Q7) This query is not clear. We investigated one study population using a single protocol.
Q8) The sentence has been rephrased.
see Introduction, Page 3, lines 75-77
Q9) Thank you, we avoided the use of since in revised manuscript.
Q10) The use of English language has been largely revised.
Introduction has been improved. In fact, additional relevant references were added and discussed in Introduction and Discussion section.
Round 2
Reviewer 3 Report
the authors have addressed the reviewer's comments. I recommend this paper to be accepted.